# Antithrombin Activity and Association with Risk of Thrombosis and Mortality in Patients with Cancer

**DOI:** 10.3390/ijms232415770

**Published:** 2022-12-12

**Authors:** Cornelia Englisch, Oliver Königsbrügge, Stephan Nopp, Florian Moik, Peter Quehenberger, Matthias Preusser, Ingrid Pabinger, Cihan Ay

**Affiliations:** 1Clinical Division of Haematology and Haemostaseology, Department of Medicine I, Medical University of Vienna, 1090 Vienna, Austria; 2Division of Oncology, Department of Internal Medicine, Medical University of Graz, 8036 Graz, Austria; 3Department of Laboratory Medicine, Medical University of Vienna, 1090 Vienna, Austria; 4Clinical Division of Oncology, Department of Medicine I, Medical University of Vienna, 1090 Vienna, Austria

**Keywords:** antithrombin, antithrombin deficiency, cancer-associated thrombosis

## Abstract

Venous and arterial thromboembolism (VTE/ATE) are common complications in cancer patients. Antithrombin deficiency is a risk factor for thrombosis in the general population, but its connection to risk of cancer-associated thrombosis is unclear. We investigated the association of antithrombin activity levels with risk of cancer-associated VTE/ATE and all-cause mortality in an observational cohort study including patients with cancer, the Vienna Cancer and Thrombosis Study. In total, 1127 patients were included (45% female, median age: 62 years). Amongst these subjects, 110 (9.7%) patients were diagnosed with VTE, 32 (2.8%) with ATE, and 563 (49.9%) died. Antithrombin was not associated with a risk of VTE (subdistribution hazard ratio (SHR): 1.00 per 1% increase in antithrombin level; 95% CI: 0.99–1.01) or ATE (SHR: 1.00; 95% CI: 0.98–1.03). However, antithrombin showed a u-shaped association with the risk of all-cause death, i.e., patients with very low but also very high levels had poorer overall survival. In the subgroup of patients with brain tumors, higher antithrombin levels were associated with ATE risk (SHR: 1.02 per 1% increase; 95% CI: 1.00–1.04) and mortality (HR: 1.01 per 1% increase; 95% CI: 1.00–1.02). Both high and low antithrombin activity was associated with the risk of death. However, no association with cancer-associated VTE and ATE across all cancer types was found, with the exception of in brain tumors.

## 1. Introduction

Venous thromboembolism (VTE) is a common complication in patients with cancer [1]. The latest studies suggest that cancer increases the risk of VTE by 9- to 15-fold compared to the general, non-cancer, population [2,3]. In addition, the risk of arterial thromboembolism (ATE) is increased in patients with cancer and is 3-fold higher than in non-cancer patients [2]. VTE and ATE are major causes of morbidity and mortality in patients with cancer [1]. VTE is a negative prognostic predictor across different cancer types [4,5,6,7]. Several biomarkers reflecting an activation of the hemostatic system, including peak thrombin generation and prothrombin fragment 1 + 2, have been found to predict VTE occurrence in patients with cancer [8,9]. However, the risk factors and mechanisms of cancer-associated thrombosis seem to be diverse and are therefore not completely understood. Some VTE risk factors in the general population, such as the factor V Leiden mutation, have also been found to contribute to VTE risk in patients with cancer [10].

Antithrombin is an important physiological anticoagulant protein, as it potently inactivates, mainly but not exclusively, thrombin and factor Xa [11]. Antithrombin deficiency, defined as values under 70% in the activity assay [12], is associated with an up to 14-times higher risk of VTE [13]. Antithrombin deficiency, although rare in the general population, is a well-known inherited risk factor of VTE and has one of the highest VTE risks among the hereditary thrombophilias [13,14,15,16]. In contrast, the association of antithrombin deficiency and ATE risk in the general population is not as strong; however, there are data suggesting that low levels may also increase ATE risk [17,18,19]. Interestingly, this was seen more prominently in female persons of a younger age [18], who also had a higher risk for recurrent events [17].

The role of antithrombin in cancer-associated thrombosis has not yet been investigated. Therefore, we aimed to investigate whether antithrombin activity levels play a role in cancer-associated VTE and ATE, and we also analyzed their association with all-cause mortality in a large prospective cohort of patients with cancer.

## 2. Results

### 2.1. Patient Characteristics

In total, 1127 patients were included in the study, of which 45% were female. The median age was 62 years (IQR: 52–68). The patients were followed for a median of 18 months (IQR: 6–24) for the primary outcomes VTE and ATE. During this period, 110 patients were diagnosed with VTE (cumulative 6-month, 12-month and 24-month incidence: 7%, 95% CI: 5.5–8.6; 9.1%, 95% CI: 9.1–10.6; 11.3%; 95% CI: 9.3–13.3), 32 with ATE (cumulative 6-month, 12-month, 24-month incidence: 1.5%, 95% CI: 0.7–2.2; 2.2%, 95% CI: 1.2–3.1; 3.7%, 95% CI: 2.4–5) and 563 patients had died (6-month, 12-month, 24-month mortality: 12.5%, 95% CI: 10.5–14.4; 26.1%, 95% CI: 23.5–28.6; 42.1%, 95% CI: 39.2–45) (Table 1, Appendix A).

### 2.2. Distribution of Antithrombin Levels

The median antithrombin activity level in the cohort was 104% (IQR 94–114%). Figure 1 shows the distribution of antithrombin levels according to tumor types. Interestingly, patients with brain tumors had the highest levels and patients with multiple myeloma had the lowest (Appendix A).

Antithrombin activity levels showed a weak negative correlation with acute phase proteins (D-dimer, fibrinogen and CRP) (Appendix A).

### 2.3. Antithrombin Activity and Risk of VTE and ATE

Across the whole cohort, there was no association between antithrombin activity and the risk of VTE (SHR: 1.01 per 1% increase, 95% CI: 0.99–1.02) or ATE (SHR: 1.00 per 1% increase, 95% CI: 0.97–1.02). In addition, after the multivariable adjustment for cancer type, stage (stage 4 versus stage 1,2 and 3), sex and age there was no association with risk of VTE (SHR: 1.00 per 1% increase, 95% CI: 0.99–1.01) and ATE (SHR: 1.00 per 1% increase, 95% CI: 0.98–1.03); Table 2.

In the subgroup of patients with antithrombin activity levels lower than 70% (*n* = 13), i.e., fulfilling the criterion for antithrombin deficiency, one patient had a VTE and one patient an ATE event. In the multivariable analysis adjusted for cancer type, stage (stage 4 versus stage 1,2 and 3), sex and age there was no association between antithrombin activity levels and VTE (SHR: 0.72 per 1% increase, 95% CI: 0.09–5.6) or ATE (SHR: 3.2 per 1% increase, 95% CI: 0.41–24.96).

In the subgroup of patients with brain tumors (*n* = 194), there was no association between antithrombin activity and risk of VTE (SHR: 0.99 per 1% increase, 95% CI: 0.96–1.03); however, higher levels were associated with an increased risk of ATE (SHR: 1.02 per 1% increase, 95% CI: 1.00–1.04). This association of antithrombin with risk of ATE remained significant upon multivariable adjustment for sex and age (SHR: 1.02 per 1% increase, 95% CI: 1.00–1.04) (Table 2).

### 2.4. Antithrombin Activity and All-Cause Mortality

We did not observe a linear association between antithrombin activity and all-cause mortality in the total study cohort. However, there was a u-shaped association between antithrombin activity levels and all-cause mortality, i.e., patients with very low levels and those with very high levels had a higher risk of mortality in the total study cohort (Figure 2, Appendix A). Patients with antithrombin levels in the 2nd and 3rd quartile (meaning antithrombin activity levels between 94–114) had a 50% lower all-cause mortality risk (HR 1.52; 95% CI: 1.26–1.82) than patients with activity levels in the 1st and 4th quartile (<94 or >114, respectively) (Appendix A).

In contrast, in patients with primary brain tumors, higher antithrombin activity levels were linearly associated with poor overall survival (HR: 1.01 per 1% increase; 95% CI: 1.00–1.02). This was also visible in the Kaplan Meier analysis when comparing antithrombin levels >75th versus <75th percentile of the distribution in brain tumor patients (6-month cumulative incidence 22.7% versus 12.9%, 12-month 50.5% versus 28.4%, and 24-month 78.6% versus 55.2%, log-rank test 0.002) (Figure 3). Primary brain tumor patients with antithrombin activity levels above the upper cut-off level of 120%, according to the definition of the routine coagulation laboratory of our institution, had a 59% higher hazard of mortality compared to those with levels below this cut-off (HR 1.59; 95% CI: 1.07–2.36) (Appendix A).

## 3. Discussion

In this prospective cohort study, we did not find an association between antithrombin activity levels and VTE and ATE in the overall cohort of patients with different types of cancer. However, we observed that in patients with brain tumors, higher antithrombin activity levels were associated with an increased risk of ATE and poor overall survival. Furthermore, cancer patients with very low or very high antithrombin levels had an increased risk of all-cause mortality in the total study cohort (u-shaped association).

Antithrombin is a potent anticoagulant protein that primarily, but not exclusively, inhibits thrombin and factor Xa [11]. Thus, an antithrombin deficiency has been linked to an increased risk of thrombotic events, including VTE and ATE, in the general population [13,17]. In previous studies, some but not all risk factors for thrombosis in the general population, including inherited thrombophilia, have also been found to be associated with the risk of cancer-associated thrombosis [1,10]. The prevalence of antithrombin deficiency in the population is low [20,21]; similarly, in our cancer population, only 13 patients fulfilled the criteria for anti-thrombin deficiency, which did not allow us to conclude on its impact on the risk of cancer-associated thrombosis. However, risk factors for VTE or ATE in the general population may be trumped by cancer- and treatment-specific risk factors in populations of cancer patients.

Patients with primary brain tumors had the highest antithrombin activity levels in our study, and only one primary brain tumor patient had an antithrombin activity value below 80%, which is the lower cut-off level according to our routine coagulation laboratory. The higher risk of ATE with increasing antithrombin levels was surprising and counterintuitive at the first look. Patients with brain tumors are among those with the highest risk of thrombosis, and in previous studies it has been observed that risk factors for thrombosis in patients with brain tumors differ from other cancer types [22,23,24,25,26]. For instance, a low platelet count was found in brain tumor patients who developed VTE, while in solid tumors, thrombocytosis is a risk factor for thrombosis [25]. A previous study also reported higher antithrombin levels in patients with brain tumors [27], but did not investigate its association with thrombosis. It is noteworthy that, in our study, higher levels were also associated with poor overall survival in this patient cohort. In contrast, the previous study found no association between antithrombin levels and overall survival [27]. The reason for these counterintuitive and opposing results remains unclear. However, we hypothesize that antithrombin could reflect changes in other components of the hemostatic system. Thus, we believe that further research investigating this issue would be of interest. Another possibility is that an alternatively spliced isoform of antithrombin that was previously described to be present in human brain tissue [28] was detected in the antithrombin activity assay.

Further, the question whether antithrombin is an acute phase parameter is relevant here, as this could be the reason for our observed association. However, there are data that suggest that it is not influenced by an acute phase reaction [29,30]. Our finding of a negative correlation between antithrombin and some acute phase proteins (CRP, fibrinogen) supports this hypothesis.

Our finding of a u-shaped association of low and high levels of antithrombin activity with an increased risk of all-cause mortality is new. Interestingly, patients with antithrombin activity levels within the normal range (80–120%) had the lowest all-cause mortality risk. This could indicate that a balanced hemostatic system has a favorable effect on the prognosis of patients. This would be in line with previous literature that reported that VTE and several biomarkers of hemostasis are negative prognostic markers in patients with different cancer types [4,5,6,7]. The association between very low levels of antithrombin with an increased all-cause mortality could suggest that very low levels are a sign of a poor performance status of patients. The reason for the association of high levels with poor overall-survival remains unclear. As antithrombin is primarily produced by the liver, one explanation could be the detection of an alternatively spliced isoform [28] that was previously found to also be present in the liver. The detailed mechanisms behind this observation remain to be elucidated.

We also adjusted for potential confounders. As the association between low antithrombin levels and the risk of ATE was reported to be more pronounced in female patients of a younger age in the general population [18], we adjusted for age and sex in multivariable analysis; however, this did not change our findings.

We would like to address some limitations of our analysis. Although the total number of patients included in this analysis (*n* = 1127) is high, the total number of patients with brain tumors (*n* = 194) is relatively low. Thus, we had a lower statistical power for the association of antithrombin activity levels with the risk of ATE and mortality in patients with brain tumors. In addition, the number of patients with decreased levels is low; therefore, we cannot exclude the possibility that a deficiency of antithrombin is a risk factor for VTE in cancer patients. Furthermore, the number of patients with very high levels was also low, which led to a wide confidence interval in our analysis. However, our study provides a new finding and opens the path to further investigations. Further studies are also needed to confirm the prognostic relevance and specifically address the predictive potential of antithrombin in patients with cancer. Previously, several hemostatic biomarkers have been shown to have a prognostic and predictive potential regarding survival and therapy response in patients with cancer. Such biomarkers could contribute to individual, personalized, risk-stratified patient management in the future [31].

To conclude, we found a u-shaped association between antithrombin activity levels and the risk of all-cause mortality in patients with cancer. In the subgroup of patients with brain tumors, an association with higher antithrombin levels and an increased ATE risk and poor overall survival was observed. There was no association found between antithrombin levels and the risk of VTE. Further studies are needed to clarify and understand the role of antithrombin in the risk of thrombosis, particularly ATE, and its association with the prognosis of patients with cancer.

## 4. Materials and Methods

### 4.1. Study Cohort

We analyzed the dataset of the Vienna Cancer and Thrombosis Study (CATS). This study is a single-center prospective observational cohort study at the Medical University of Vienna. The study has been approved by the local ethics committee (Ethics Committee of the Medical University of Vienna; EC number: 126/2003, ethik-kom@meduniwien.ac.at) and was performed according to the Declaration of Helsinki. The study included patients with newly diagnosed or recurrent cancer, who provided written informed consent. This study was originally designed to elucidate the risk factors for cancer-associated VTE and followed patients for 2 years for the primary outcome VTE. Secondary outcomes included death and ATE [32]. More detailed information about the study has been published previously [8,33]. The exclusion criteria included an indication for long-term anticoagulation, overt viral or bacterial infection, radiotherapy, or surgery in the last 2 weeks and chemotherapy or VTE in the 3 months prior to study inclusion and sample collection.

For VTE confirmation objective imaging, methods such as duplex/compression sonography or venography for DVT and computed tomography or ventilation/perfusion lung scan for pulmonary embolism (PE) were used. Symptomatic ATE, which was defined as a composite of acute myocardial infarction (ST-elevation myocardial infarction and non-ST-elevation myocardial infarction), peripheral arterial occlusion, if treated with an interventional procedure (i.e., a catheter-based or open surgical procedure to improve arterial blood flow in non-cardiac arteries, except the intracranial vessels), and ischemic stroke were diagnosed with: (1) computed tomography (CT), magnetic resonance imaging (MRI) and autopsy report for ischemic stroke; (2) Doppler-sonography, digital subtraction angiography, CT-angiography, and MR-angiography for peripheral arterial occlusion; (3) echocardiography (e.g., hypokinetic/akinetic and hypotrophic myocardial section without any other existing reason), cardiac biomarkers, identification of an intracoronary thrombus by angiography, and autopsy evidence for myocardial infarction [16]. An adjudication committee, comprising independent specialists in vascular medicine (angiology), radiology, nuclear medicine, cardiology or neurology verified all thrombotic events. Asymptomatic arterial thrombosis (e.g., incidentally detected stroke on restaging CT scans) was considered an event if it was considered clinically significant by members of the adjudication committee.

For the current analysis, we used data from 1127 patients for whom antithrombin activity levels measured at study inclusion were available. The antithrombin activity levels were assessed with a chromogenic test [34,35,36] in the General Hospital Vienna within the routine clinical care in the Department of Laboratory Medicine. The STA^®^ Stachrom^®^ AT III 3 Assay (Diagnostica stago, Asnières-sur-Seine, France) was used.

At the Department of Laboratory Medicine of the Medical University of Vienna, the normal range of antithrombin activity is 80–120%. Values below 70% are considered to fulfill the criterion of antithrombin deficiency [12].

### 4.2. Statistical Analysis

The statistical analyses were performed with SPSS 28.0 (IBM SPSS Statistics, Chicago, IL, USA), STATA 17 (Stata Corp., Houston, TX, USA) and RStudio (Boston, MS, USA). Standard summary statistics were used to report patient baseline characteristics (absolute frequencies, percentages, median, interquartile range [IQR]).

VTE and ATE outcomes were studied in a competing risk framework, as death was considered a competing event during follow-up time. Therefore, a proportional sub-hazard regression model, according to Fine and Gray, to compare the cumulative VTE and ATE incidence between groups was conducted. Cox regression was performed to investigate the association between levels of antithrombin activity and all-cause mortality. Further, restricted cubic splines were used to investigate the potential non-linear relationships between antithrombin levels and the outcome. Kaplan-Meier curves were plotted to compare overall survival between groups. A two-sided *p*-value < 0.05 was defined as threshold for statistical significance.

## Figures and Tables

**Figure 1 ijms-23-15770-f001:**
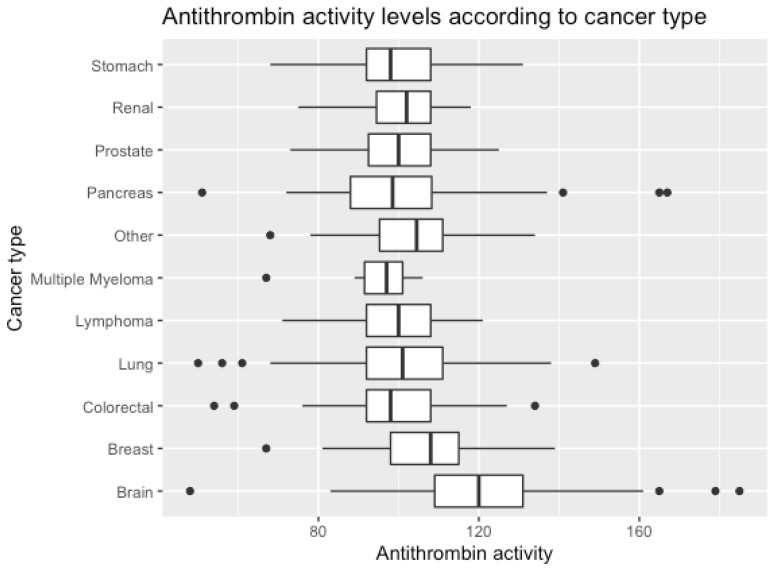
Antithrombin activity levels according to different cancer types. Category brain includes only primary brain tumors: glioma, medulloblastoma, meningioma. Category other includes esophageal, sarcoma, testis, hepatocellular, thymus, genitourinary, thyroidal, mesothelioma. Bold line represents median; upper and lower hinge represent third and first quartile, respectively; points indicate outliers.

**Figure 2 ijms-23-15770-f002:**
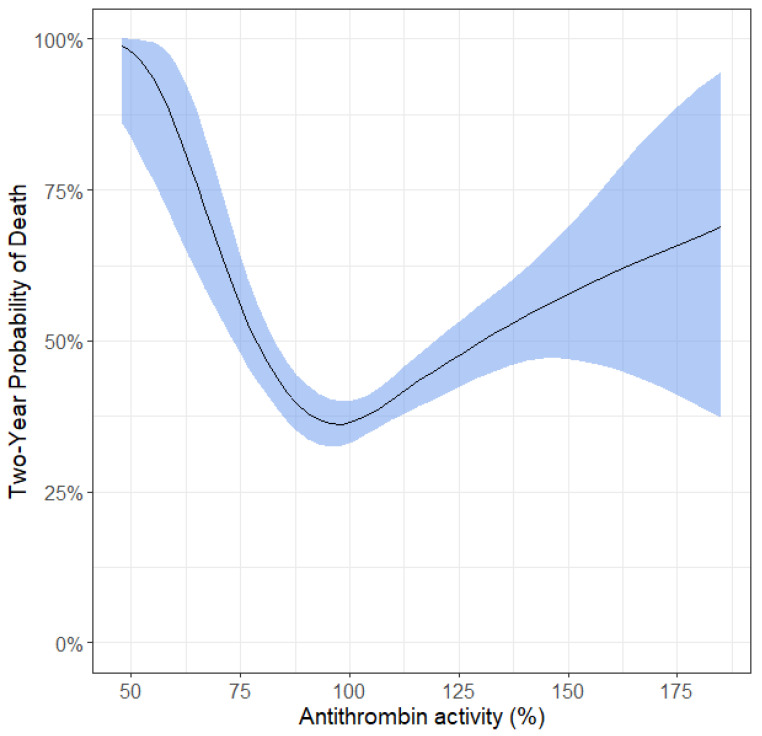
Probability of death at 2 years of follow-up in relation to antithrombin activity levels. Y axis depicts the two-year probability of death in %, i.e., a patient with 100% antithrombin activity had a probability of death at 2 years of 36%. The blue shaded area indicates 95% confidence intervals.

**Figure 3 ijms-23-15770-f003:**
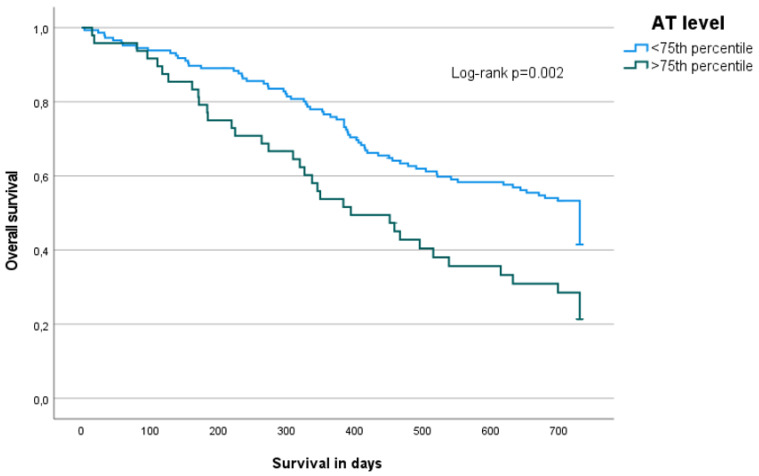
Overall-survival of brain tumor patients (*n* = 194) with antithrombin activity levels <75th (*n* = 47) (≤131.25%) versus >75th percentile (*n* = 147) (>131.25%). Patients were divided according to their antithrombin activity level and the group with levels under 131.25% (<75th percentile) was compared to the group with levels over 131.25% (>75th percentile) within a Kaplan Meier analysis and with a log-rank test *p* = 0.002.

**Table 1 ijms-23-15770-t001:** Patient characteristics at study baseline. IQR, interquartile range. SD, standard deviation.

Patient Characteristics				
	*n* (% Missing Value)	*n* (%)	Median (IQR)	Mean (+/− SD)
Age	1127 (0%)		62 (52–68)	
Female	1127 (0%)	509 (45%)		
VTE	1127 (0%)	110 (9.8%)		
ATE	1127 (0%)	32 (2.8%)		
Died	1127 (0%)	563 (50%)		
Metastatic disease (solid tumor patients)	892 (0%)	548 (61%)		
Non-classifiable (brain, hematological)	235 (0%)			
Median follow-up period (months)	1127 (0%)		18 (6–24)	
Antithrombin activity level (%)	1127 (0%)			
Whole cohort	1127 (0%)			104.9 (88.3–121.5)
Brain tumor	194 (0%)			121.2 (102.4–140)
Other	943 (0%)			101.7 (87.6–115.8)
Outcomes within 2 years follow-up	1127 (0%)			
VTE		110 (9.8%)		
ATE		32 (2.8%)		
Death		563 (50%)		

**Table 2 ijms-23-15770-t002:** Association of antithrombin levels with VTE, ATE and all-cause mortality. Results of univariable analysis and adjusted for cancer entity, stage 4 versus other, sex, and age; in patients with brain tumors adjusted for age and sex only.

	Univariable (95% CI)	*p*	Multivariable (95% CI)	*p*
Full cohort (*n* = 1127)				
Risk of VTE	1.01 (0.99–1.02)	0.343	1.00 (0.99–1.01)	0.903
Risk of ATE	1.00 (0.97–1.02)	0.743	1.00 (0.98–1.03)	0.952
Risk of mortality	1.00 (1.00–1.01)	0.608	1.00 (0.99–1.00)	0.102
Only brain tumors (*n* = 194)				
Risk of VTE	0.99 (0.96–1.03)	0.680	0.99 (0.96–1.03)	0.690
Risk of ATE	1.02 (1.00–1.04)	0.017	1.02 (1.00–1.04)	0.055
Risk of mortality	1.01 (1.00–1.02)	0.030	1.01 (1.00–1.02)	0.083
Non-brain cancer (*n* = 977)				
Risk of VTE	1.00 (0.99–1.02)	0.460	1.01 (1.00–1.02)	0.246
Risk of ATE	0.98 (0.95–1.01)	0.217	0.99 (0.96–1.02)	0.527
Risk of mortality	1.00 (0.99–1.00)	0.546	1.00 (0.99–1.00)	0.449

## Data Availability

The data presented in this study are available on request from the corresponding author. The data are not publicly available due to ethical reasons.

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
