# Peer review of "Antithrombin Activity and Association with Risk of Thrombosis and Mortality in Patients with Cancer"

_ijms, 2022, doi:10.3390/ijms232415770_

Round 1

Reviewer 1 Report

The authors present a prospective cohort study investigating the association of antithrombin levels with VTE and ATE in cancer patients.  The overall cohort was significant in size (n=1,127), albeit comparatively small for patients with brain tumors (n=194), which is where one of the two main interesting findings was observed. 

The finding of a "u"-shaped association between antithrombin levels and survival is intriguing, although the wide 95% confidence limits at high antithrombin levels indicate a lack of robustness of the data in this region, which the authors acknowledge.  

The second finding is that patients with brain tumors that have elevated antithrombin levels have an elevated risk of ATE.  Due to the low number of patients in this population, the effect is small with a hazard ratio of only 1.02, and is arguably only marginally statistically significant (p=0.055).

Having said this, it is a prosepective study and the observation is certainly intriguing and likely to be of interest to readers as grist for future studies.

Recommendations for making the data more interpretable include:

1. Use Means +/- SD in Table 1 (as opposed to Medians).
2. For the patients who died (563 as listed in Table 1), it may be useful for readers to have a breakout of those that had VTE or ATE. 
3. Change the presentation of the data in Table 2 to a Box-Whisker Plot.

Author Response

The authors present a prospective cohort study investigating the association of antithrombin levels with VTE and ATE in cancer patients.  The overall cohort was significant in size (n=1,127), albeit comparatively small for patients with brain tumors (n=194), which is where one of the two main interesting findings was observed.

The finding of a "u"-shaped association between antithrombin levels and survival is intriguing, although the wide 95% confidence limits at high antithrombin levels indicate a lack of robustness of the data in this region, which the authors acknowledge. 

The second finding is that patients with brain tumors that have elevated antithrombin levels have an elevated risk of ATE. Due to the low number of patients in this population, the effect is small with a hazard ratio of only 1.02, and is arguably only marginally statistically significant (p=0.055).

Having said this, it is a prosepective study and the observation is certainly intriguing and likely to be of interest to readers as grist for future studies.

Author response: We thank the reviewer for the thorough evaluation of our manuscript and are glad the reviewer finds our study of interest. We diligently considered all raised issues by the reviewer and put large effort into considering all raised issues appropriately. Based on that, we have made several modifications to the revised version of our manuscript. Below, find our detailed responses and explanations on all raised suggestions and remarks in a point-by-point fashion.

Recommendations for making the data more interpretable include:

  1. Use Means +/- SD in Table 1 (as opposed to Medians).

Author response: We thank the reviewer for her/his comment. According to standard reporting we have chosen to show our results as median with IQR as most are not normally distributed and the median is not as heavily influenced as the mean by outliers. We agree that normally distributed variables should be reported as means +/- SD. Therefore, we added the means +/- SD for the normally distributed variables in Table 1 (page 2).

  1. For the patients who died (563 as listed in Table 1), it may be useful for readers to have a breakout of those that had VTE or ATE.

Author response: We agree with the reviewer that these data would be of interest to readers. We added this information in form of a table in the supplement (supplementary table 5).

  1. Change the presentation of the data in Table 2 to a Box-Whisker Plot.

Author response: We thank the reviewer for this suggestion. We converted table 2 into a Boxplot and put table 2 into the supplement. Page 3, Figure 1

Reviewer 2 Report

Cornelia et al suggested that Antithrombin activity shows a u-shaped association with the risk of death in patients with cancer but is not associated with cancer-associated VTE and ATE across all cancer types. This study is interesting, results from this study provide new information on cancer and CV system. However, there are still several issues that need to be improved. A revision is suggested.

1 Please strengthen the conclusion in the abstract.

2 Although cancer-associated thrombosis has not been fully studied. In the introduction, please emphasize the role of thrombosis or thrombin activity in cancer or cancer treatments.

3. Please discuss the limitation of this study.

4. Please address the clinical implications of this study, and how to apply the findings from this study to a clinical survey.

Author Response

Cornelia et al suggested that Antithrombin activity shows a u-shaped association with the risk of death in patients with cancer but is not associated with cancer-associated VTE and ATE across all cancer types. This study is interesting, results from this study provide new information on cancer and CV system. However, there are still several issues that need to be improved. A revision is suggested.

Author response: We thank the reviewer for the thorough evaluation of our study and the important comments provided. We have put much effort into elaborating on all raised issues and suggestions, and in revising the manuscript accordingly. The reviewer´s feedback has helped greatly in further improving our manuscript. Please find below the answers to the comments in a point-by-point fashion below.

1. Please strengthen the conclusion in the abstract.

Author response: We thank the reviewer for this suggestion. We improved our conclusion in the abstract.
            Page 1, lines 24-26 “Both high and low antithrombin activity was associated with risk of death. However, no association with cancer-associated VTE and ATE across all cancer types was found, except in brain tumors.”

2. Although cancer-associated thrombosis has not been fully studied. In the introduction, please emphasize the role of thrombosis or thrombin activity in cancer or cancer treatments.

Author response: We agree with the reviewer that this is a very important issue. We carefully revised our introduction and put more emphasis on the importance of thrombosis and thrombin generation in patients with cancer.
            Page 1, lines 35-38 “VTE is a negative prognostic predictor across different cancer types.(4-7) Several biomarkers reflecting an activation of the hemostatic system including peak thrombin generation and prothrombin fragment 1+2 have been found to predict VTE occurrence in patients with cancer.(8, 9)”

  1. Please discuss the limitation of this study.

Author response: We thank the reviewer for his/her comment. We cautiously reviewed our limitations section in the discussion of our manuscript and made adjustments.
            Page 7, lines 235-238 “. Furthermore, the number of patients with very high levels was low as well, which led to a wide confidence interval in our analysis.”

  1. Please address the clinical implications of this study, and how to apply the findings from this study to a clinical survey.

Author response: We appreciate the reviewer’s thorough evaluation of our manuscript. Based on our data, it is difficult for us to draw a specific clinical implication. Further studies are needed to confirm the prognostic relevance of antithrombin and specifically address its potential to predict risk of mortality. However, once this is confirmed, antithrombin could be a useful prognostic marker. Previously, several hemostatic biomarkers have been shown to have a prognostic and predictive potential regarding survival and therapy response in patients with cancer. Such biomarkers could contribute to individual, personalized, risk-stratified patient management in the future (see Moik F and Ay C, JTH 2022: Hemostasis and cancer: Impact of haemostatic biomarkers for the prediction of clinical outcomes in patients with cancer).
We hope the reviewer understands our thought-process and our reasoning regarding this matter. We added a statement regarding this in the revised discussion section of our manuscript.
            Page 7, lines 239-244 “Further studies are also needed to confirm the prognostic relevance and specifically address the predictive potential of antithrombin in patients with cancer. Previously, several hemostatic biomarkers have been shown to have a prognostic and predictive potential regarding survival and therapy response in patients with cancer. Such bi-omarkers could contribute to individual, personalized, risk-stratified patient manage-ment in the future.(31)”

Reviewer 3 Report

The authors included large number of patients suffering from cancer and concomitant thromboembolism, ATE and cancer In their analysis only subjects with brain tumor had higher risk of ATE and antithrombin activity. Furthermore, this is clinically more relevant in terms of mortality. Yet, they fail to hypothesize the pathophysiological mechanism of these finding. I would only recommend to expand this part of discussion. The study is plausible and after minor revisions, the manuscript could be accepted.  

Author Response

The authors included large number of patients suffering from cancer and concomitant thromboembolism, ATE and cancer In their analysis only subjects with brain tumor had higher risk of ATE and antithrombin activity. Furthermore, this is clinically more relevant in terms of mortality. Yet, they fail to hypothesize the pathophysiological mechanism of these finding. I would only recommend to expand this part of discussion. The study is plausible and after minor revisions, the manuscript could be accepted.

Author response: We thank the reviewer for the thorough evaluation of our manuscript and the comments provided. We revised our discussion carefully and further slightly expanded our hypothesis about the pathophysiological mechanism of our findings.
            Page 6, lines 217-219 “This would be in line with previous literature that reported that VTE and several biomarkers of hemostasis are negative prognostic markers in patients with different cancer types.(4-7)”
            Page 7, lines 223-225 “As antithrombin is primarily produced by the liver, one explanation could be the de-tection of an alternatively spliced isoform(22) that was previously found to be also present in the liver.”

Round 2

Reviewer 2 Report

My questions had been well addressed. This study is acceptable with this vision.